# Pathological Significance of Macrophages in Erectile Dysfunction Including Peyronie’s Disease

**DOI:** 10.3390/biomedicines9111658

**Published:** 2021-11-10

**Authors:** Yasuyoshi Miyata, Tomohiro Matsuo, Yuichiro Nakamura, Kensuke Mitsunari, Kojiro Ohba, Hideki Sakai

**Affiliations:** Department of Urology, Graduate School of Biomedical Sciences, Nagasaki University, Nagasaki 852-8501, Japan; yasu-myt@nagasaki-u.ac.jp (Y.M.); yn1238056@yahoo.co.jp (Y.N.); ken.mitsunari@gmail.com (K.M.); ohba-k@nagasaki-u.ac.jp (K.O.); hsakai@nagasaki-u.ac.jp (H.S.)

**Keywords:** erectile dysfunction, Peyronie’s disease, phosphodiesterase type 5 inhibitor, macrophage

## Abstract

Erectile function is regulated by complex mechanisms centered on vascular- and nerve-related systems. Hence, dysregulation of these systems leads to erectile dysfunction (ED), which causes mental distress and decreases the quality of life of patients and their partners. At the molecular level, many factors, such as fibrosis, lipid metabolism abnormalities, the immune system, and stem cells, play crucial roles in the etiology and development of ED. Although phosphodiesterase type 5 (PDE5) inhibitors are currently the standard treatment agents for patients with ED, they are effective only in a subgroup of patients. Therefore, further insight into the pathological mechanism underlying ED is needed to discuss ED treatment strategies. In this review, we focused on the biological and pathological significance of macrophages in ED because the interaction of macrophages with ED-related mechanisms have not been well explored, despite their important roles in vasculogenic and neurogenic diseases. Furthermore, we examined the pathological significance of macrophages in Peyronie’s disease (PD), a cause of ED characterized by penile deformation (visible curvature) during erection and pain. Although microinjury and the subsequent abnormal healing process of the tunica albuginea are known to be important processes in this disease, the detailed etiology and pathophysiology of PD are not fully understood. This is the first review on the pathological role of macrophages in PD.

## 1. Introduction

Erectile response in males is suggested to be a series of five phases: latent, tumescent, full erection, rigid erection, and detumescent [1]. Various factors, including central and peripheral neural signaling, smooth muscle contraction and relaxation, and blood flow in the corpus cavernosum, are associated with erectile function via complex mechanisms. Specifically, sexual arousal stimulates the release of nitric oxide (NO) at the nerve endings and in vascular endothelial cells in the penis. Subsequently, NO increases the production of cyclic guanosine monophosphate (cGMP) from guanosine triphosphate (GTP) by activating guanyl cyclase in the cavernous smooth muscle cells of the corpus cavernosum. Furthermore, increased cGMP activates cGMP-dependent protein kinase (PKG), and the subsequent phosphorylation of various proteins, resulting in decreased intracellular calcium levels and a consequent relaxation of smooth muscles, leading to penile erection [2,3]. In addition to these factors, biomolecular activities, such as prostaglandins and oxidative stress, play important roles in the cGMP-related pathway in erectile function [1,4].

Erectile dysfunction (ED) affects the quality of life in men because it causes difficulties with vaginal penetration during intercourse; this may also lead to mental distress. Additionally, ED often decreases the quality of life of the partner of patients with ED. In recent years, ED has been reported to be a major cause of depression in patients with chronic spinal cord injury [5]. Therefore, the development of treatment strategies for ED is important, and understanding the underlying mechanism is essential. ED occurs when intracellular cGMP levels decrease due to the conversion of cGMP to 5′-guanosine monophosphate (5′-GMP) by phosphodiesterase type 5 (PDE5) [2,6]. The level of cGMP in smooth muscle cells plays a crucial role in penile erection, which is regulated by the balance between cGMP synthesis by guanyl cyclase and cGMP hydrolysis by PDE5 [2,3,6]. Therefore, PDE5 inhibitors, which increase cGMP levels, are useful for the treatment of ED. In fact, the current standard therapy for ED is an oral medication of PDE5 inhibitors, such as sildenafil, tadalafil, and vardenafil. However, PDE5 inhibitors are effective only for a subgroup of ED patients [7]. Apart from molecular mechanisms that are centered on cGMP, many other factors have also been recognized to cause ED, such as vascular disorders, trauma- or operation-induced nerve injury, and inflammatory disease of the penis [8,9,10,11,12]. Therefore, novel therapeutic agents for ED need to be developed, and more detailed and wider knowledge is necessary to address this issue.

Various factors, including vasculogenic, neurogenic, physiologic, endocrinologic, and drug-related factors, have been suggested to be involved in the underlying pathogenesis of ED, and erectile function is often affected by several factors [13]. Furthermore, there is a general agreement that monocytes/macrophages play a crucial role in vascular diseases [14,15]. Moreover, macrophages are suggested to play significant pathological roles in the etiology and development of non-vascular-related ED. However, unfortunately, there are limited reviews on the pathological roles of macrophages in ED, especially in neurogenic ED and penile fibrosis-related ED. Therefore, in this review, we focused on the role of macrophage accumulation and activity in ED pathogenesis. In addition, this is the first review on the pathological functions of macrophages in Peyronie’s disease (PD), a cause of ED. Moreover, we discuss the pharmacological effects of macrophages in the treatment of ED.

## 2. Vasculogenic Erectile Dysfunction

While discussing the pathological mechanisms of vasculogenic ED, it is important to discuss the endothelial dysfunction caused by lipid metabolism disorder in blood vessels [16,17]. In particular, low-density lipoprotein (LDL) abnormalities are a well-known key factor in the development of endothelial dysfunction in various vascular diseases, such as atherosclerosis [18,19]. Furthermore, macrophages play an important role in abnormal LDL-related vascular damage by stimulating plaque formation [19]. Oxidized LDL (ox-LDL) promotes atherosclerosis and increases the risk of cardiovascular diseases [20,21]. Moreover, ox-LDL can affect the activities and phenotypes of macrophages by regulating cytokine production, inflammation, and lipid uptake [15,21,22,23]. Thus, the interaction between ox-LDL and macrophages is a key factor in the progression of endothelial injury. A simple schematic of the pathological roles of ox-LDL and macrophages in endothelial dysfunction and vascular damage is shown in Figure 1.

As mentioned above, ED is recognized as a vascular disease. In fact, various vascular diseases, such as atherosclerosis, coronary heart disease, and peripheral artery disease, are known to cause ED because endothelial damage and subsequent atherosclerotic plaque formation are common etiologies of ED and other vascular diseases [24,25]. Lipid metabolism abnormalities and monocyte/macrophage activity are reported to be crucial factors in the relevant pathological steps, including atherosclerotic plaque formation [26,27,28,29,30]. Thus, the interaction between lipid metabolism and macrophages is speculated to be closely associated with the pathogenesis and development of ED. However, few studies have been conducted on the pathological role of the interaction between ox-LDL and macrophages in ED. Myeloperoxidase (MPO)-dependent ox-LDL (Mox-LDL) was detected in penile tissues in patients with vascular ED, but not in those with prostatectomy-caused neurogenic ED [31]. Importantly, macrophage-derived MPO promotes endothelial dysfunction and plaque instability [32,33]. Based on these facts, macrophages have been suggested to significantly influence endothelial conditions via Mox-LDL [34]. In contrast, other investigators have shown that Mox-LDL stimulates macrophage functions, including reactive oxygen species (ROS) production and cytokine secretion, which are closely associated with endothelial conditions [8,35,36]. Unfortunately, the detailed pathological roles of the interaction between macrophages and Mox-LDL in ED are unclear. However, because Mox-LDL is present in penile tissues in ED and macrophages play a crucial role in endothelial dysfunction and atherosclerotic plaque formation in cooperation with Mox-LDL [8], macrophages are speculated to be key players in vasculogenic ED via the modulation of lipid metabolism and endothelial conditions.

## 3. Neurogenic Erectile Dysfunction

In addition to vasculogenic factors, neurogenic system dysfunction is a major cause of ED. Although various factors have been reported to be etiological factors of neurogenic ED, major surgery and radiotherapy of the pelvis/retroperitoneum are well-known causes of neurogenic ED [37]. In fact, ED is one of the most common sequelae of radical prostatectomy and radiation therapy for prostate cancer [10,12]. With the development of new surgical techniques and devices, including robot-assisted radical prostatectomy with nerve-sparing techniques, modern external beam radiotherapy, brachytherapy, rehabilitation, and treatment options for ED, the frequency and degree of prostatectomy-induced ED is decreasing [12,38,39,40]. However, after such local treatments, ED is unavoidable to a certain extent. The frequency of ED after radical prostatectomy has been reported to be 75–80% [41]. Damage of the penile neurovascular bundle and cavernous nerves, which causes reduced oxygenation and structural changes of the tissue in the corpora cavernosa, is suggested to be the etiology of this ED type [42]. Therefore, ED caused by neurological disorders is classified as neurogenic ED and is distinguished from vascular ED based on etiological and pathological process differences. In addition to nerve injury, various pathological conditions can cause neurogenic ED [37]. For example, degenerative disorders, spinal cord trauma, and stroke are central causes, whereas diabetes mellitus, chronic renal failure, and neuropathy are peripheral causes [37]. Moreover, some individuals with ED have overlapping pathological characteristics due to neurogenic and vasculogenic etiology. 

A variety of cytokines have been reported to be significantly associated with the etiology of ED due to cavernous nerve injury in animal models [42,43]. For example, compared to sham-operated rats, a rat ED model demonstrated a higher mRNA expression of both pro-inflammatory (interleukin (IL)-1b, IL-6, tumor necrosis factor (TNF)-α, transforming growth factor (TGF)-β, C-C motif chemokine ligand (CCL) 2, and CC chemokine receptor (CCR) 2) and anti-inflammatory (IL-10) cytokines in the major pelvic ganglion 48 h after bilateral cavernous nerve injury [42]. Moreover, the levels of M1 markers (CD86, inducible nitric oxide synthase, interleukin-1β) and M2 markers (CD206, arginase-1, and interleukin-10) increased after bilateral cavernous nerve injury in the major pelvic ganglion. However, although inducible NO synthase expression increased, arginase-1 expression remained unchanged. In addition, immunohistochemical analysis showed that M1 (CD68 + CD86 +) macrophages increased predominantly in the major pelvic ganglion after bilateral cavernous nerve injury [42]. Based on these findings, it was suggested that the recruitment of M1 macrophages into the major pelvic ganglion after bilateral cavernous nerve injury was higher than that of M2 macrophages, and that such increased levels of M1 macrophages were associated with neurogenically impaired erectile tissue physiology [42]. Furthermore, the same research group reported that bilateral cavernous nerve injury increased TNF-α expression in the major pelvic ganglion, which suppressed major pelvic ganglion neurite outgrowth [43]. Finally, the authors suggested that TNF-α inhibition may prevent ED caused by cavernous nerve injury and loss of nitrergic nerve apoptosis and preserve corporal smooth muscle function. Importantly, they speculated that M1 macrophages play a significant role in this mechanism [43].

In addition to TNF-α, TGF-β is also recognized as a key factor in the development of ED [26,44]. It is known that TGF-β and macrophages cooperate with each other under many physiological and pathological conditions [45,46,47]. Hence, it is possible that the interaction between macrophages and TGF-β is essential for the development of ED. However, the detailed pathological role of such interactions in ED remains unclear. Nevertheless, it has been reported that the density and location of macrophages, detected using F4/80 staining, in Tgfb1−/− mice were similar to those in Tgfb1+/+ mice, although F4/80-stained macrophages were located in the dermal tissues beneath the penile spines and the vasculature of the erectile tissues [48]. Thus, further in vivo and in vitro studies are necessary to clarify the pathological roles of this interaction.

## 4. Obesity-Related Erectile Dysfunction and Macrophages

### 4.1. High-Fat Diet and Erectile Dysfunction

Lifestyle and diet are known to be closely associated with the pathogenesis of ED, especially in vasculogenic ED [49]. Diabetes mellitus caused by a high-calorie and high-fat diet is a representative cause of neurogenic ED [37]. Furthermore, a high-fat diet leads to coronary artery disease and benign prostatic hyperplasia, which are associated with ED symptoms [50,51]. Hence, the pathogenetic mechanisms of high-fat diet-induced ED, with attendant obesity, have been the research focus of many investigators. However, it remains unclear how high-fat diet and obesity affect erectile function and structural changes in penile tissues. In recent years, obesity mouse models have shown that a high-fat diet impairs pre-penile or penile smooth muscle vasoreactivity and penile neurotransmitter-mediated relaxation [52]. Furthermore, a combination of a high-fat diet and a variety of drugs has been reported to promote the development of corpora cavernosa fibrosis in a mouse model [53]. Thus, although obesity and high-fat diet are speculated to be closely associated with pathological changes in penile tissues, more detailed studies on the influence of these factors on ED at the molecular level are necessary to discuss the prevention and treatment strategies of obesity- and high-fat diet-induced ED. Moreover, the pathological roles of macrophages in obesity and high-fat diet-related ED are not fully understood.

Mineralocorticoid antagonists have been reported to show vasculoprotective effects in a murine model of mild cholesterol-induced atherosclerosis [54]. The suggested underlying mechanisms include reduction in oxidative stress, atherosclerotic lesion size, fibrosis, and macrophage count in atherosclerotic plaque lesions. Interestingly, a decrease in collagen content was detected in the penile corpora cavernosa and sinus aorticus of mice treated with mineralocorticoid antagonists. Based on these findings, it was suggested that mineralocorticoid antagonists had beneficial effects on ED by improving penile endothelial function [54]. However, this study did not elucidate the pathological roles of macrophages in ED.

### 4.2. Cytokines and Macrophages in High-Fat Diet-Related Erectile Dysfunction

A previous study reported that a high-fat diet increased TNF-α levels in the serum and corpus cavernosum of wild-type mice; however, such changes were not found in toll-like receptor (TLR)9 mutant mice [55]. Importantly, although TNF-α was produced by macrophages through the activation of TLR9 in wild-type mice, macrophage-related TNF-α production was not detected in TLR9 mutant mice [55]. Briefly, the results of the study suggested that obesity induced by a high-fat diet led to systemic inflammation and increased the expression of TNF-α in the corpus cavernosum, which was in part mediated by the activation of TLR9 expressed in macrophages [55]. Thus, the authors suggested that the innate immune system is a potential therapeutic target for the treatment of erectile complications in obesity [55]. However, the authors mentioned, as one of their study limitations, that these examinations were performed only in vitro using isolated tissues. Thus, there are no data on the in vivo response of TLR9 in macrophages.

Several studies have shown that macrophages play important roles in high-fat diet- and obesity-induced ED [55,56]. The immune system, including macrophage activity, is speculated to be a therapeutic target for some individuals with ED [55]. However, to elucidate these issues, further detailed in vivo and in vitro studies are necessary. Understandably, clinical trials aiming to clarify the changes in the biological and pathological roles of macrophages in penile tissues induced by a high-fat diet in human subjects are difficult to conduct because of ethical concerns.

### 4.3. Physical Activity

Numerous studies and reviews suggest that increasing physical activity through exercise and training, to create a negative energy balance, is useful for weight loss and maintenance [57,58,59,60]. In addition, physical exercise (PE) improves lipid metabolism [61,62]. As a result, PE is recognized as a major treatment strategy in various obesity-related diseases, including coronary heart disease, diabetes mellitus, fatty liver diseases, and ED [62,63,64,65]. In this subsection, we introduce the relationship between PE and macrophages in ED.

In an animal metabolic syndrome model in which rabbits were fed a high-fat diet, physical activity (12-week period of running on a treadmill) improved erectile function by restoring the alterations in the penile, testicular, and hypothalamic districts [56]. Briefly, PE increased the mRNA expression of testicular steroidogenesis-related enzymes, including HSD17B2, HSD17B3, and SRD5A1, anti-inflammatory changes in the testis and hypothalamus, and the circulating luteinizing hormone, cholesterol, and the testosterone/androstenedione ratio. Importantly, this study showed that the level of RAM11, a specific macrophage marker, in the interstitial space of the testis of rabbits treated with HFD was significantly decreased by PE; however, there was no significant difference in histological findings using hematoxylin-eosin staining [56]. In addition, the authors found that PE reversed the increased expression of macrophage-related genes, including CD68, CD11C, CD206, C-C chemokine receptor type 2, and TNF-α [56]. Thus, PE can improve metabolic syndrome-induced ED and the suppression of inflammation, and the activity of macrophages may be associated with the underlying mechanisms. Hence, this knowledge is very important for understanding the molecular mechanisms underlying the usefulness of PE for the prevention of ED in obese men.

Macrophages may affect the pathogenesis of ED via complex mechanisms; however, the detailed molecular significance of such macrophage-related activities in penile tissues is not fully understood. In Table 1, we summarize the pathological roles of macrophages in ED.

## 5. Stem Cell Therapy and Macrophages

### 5.1. Stem Cells and Erectile Dysfunction

In recent years, regenerative therapeutic strategies using various types of stem cells have attracted attention in many pathological conditions. Among the most important characteristics of stem cells is that they can regenerate and restore normal cell activity according to internal and external stimuli. For example, a variety of stem cells can differentiate into neurons, endothelial, and smooth muscle cells [66,67]. The efficacy and safety of stem cell-related ED therapies have been investigated in phase 1–2 clinical trials [68,69]. In addition, combinations of such regenerative therapies and other therapies, such as low-intensity shock wave and platelet-rich plasma therapy, may have an additive or synergistic benefit greater than any one therapy alone [70]. Thus, stem cell-based treatment strategies are expected to be promising in the near future. Interestingly, macrophages are recognized as key modulators of the stem cell-related regeneration of various tissues [71,72,73]. Therefore, in this subsection, we introduce the biological and pharmacological roles of macrophages in stem cell therapy for ED.

### 5.2. Adipose Tissue-Derived Stem Cell Therapy and Macrophages

The usefulness of adipose-tissue-derived stem cells in the treatment of ED has been suggested in various in vivo studies [74,75,76,77]. Unfortunately, the direct correlations between adipose tissue-derived stem cells and macrophages in ED are not clear. However, adipose tissue-derived stem cells have been reported to play crucial roles in macrophage activity under various pathological conditions, such as inflammation, endothelial cell damage, and obesity [78,79]. Importantly, these pathological conditions are closely associated with ED pathogenesis [16,17]. Thus, it is possible that macrophages are associated with some of the pharmacological effects of adipose tissue-derived stem cells in ED.

In vitro studies have shown that several ligand/chemokine receptor pairs and cytokines, such as CCL2/CCR4, CX3C chemokine ligand (CX3CL)1/CX3C chemokine receptor (CX3CR)1, XCL1/XCR1, macrophage-1, ILs, and TNF, play crucial roles in the regulation of adipose tissue-derived stem cell activities after nerve injury [80,81]. Interestingly, these factors are also associated with the biological characteristics and activity of macrophages under various physiological and pathological conditions [82,83,84]. Matsui et al. [42] investigated the influence of bilateral cavernous nerve injury on macrophage recruitment to the major pelvic ganglion in a rat model of ED. The authors found that the number of positively stained cells for CD68, a general macrophage marker, was increased in the major pelvic ganglion following cavernous nerve injury. Importantly, such an increase in CD68-positive cells occurred 48 h after the injury, and the maximal increase in the M1 macrophage/M2 macrophage ratio was observed 48 h following the injury. In addition, the M1 macrophage/M2 macrophage ratio was consistently above 1.0 after 48 h, 7 days, and 14 days after surgery; the ratio in the control sham-operated rats was consistently below 1.0. Based on these results, the authors suggested that bilateral cavernous nerve injury altered the polarization of macrophages to M1 macrophages. M1 macrophages are recognized as classically activated macrophages, whereas M2 macrophages act as alternatively activated macrophages. The former have harmful effects on nerve regeneration, whereas the latter have neuroprotective effects against nerve injury [85,86]. Increased infiltration of M1 macrophages into the major pelvic ganglion is speculated to play an important role in neurogenic ED due to cavernous nerve injury. Hence, suppressing the recruitment of neurotoxic M1 macrophages to the pelvic ganglion is suggested as a promising preventive strategy in neurogenic ED caused by radical prostatectomy or radiation therapy in patients with prostate cancer [42].

### 5.3. Mesenchymal Stem Cell Therapy and Macrophages

In addition to adipose tissue-derived stem cells, several in vivo studies have shown that mesenchymal stem cells (MSCs) have preventive and therapeutic effects in ED, especially in cavernous nerve injury-related ED [87,88,89]. However, the detailed molecular mechanism of the protective function of MSCs in ED has not been fully understood. In recent years, human gingiva-derived MSCs have been reported to act as protective factors for cavernous nerve injury-related ED via regulation of atrophy, apoptosis, and fibrosis in the penile tissues of a rat model [90]. The same study also showed that the skew of macrophage polarity was closely associated with anti-ED function. Briefly, the study demonstrated that injecting human gingiva-derived MSCs significantly reduced the expression of M1 macrophage markers (CD11b, CD68, and CD86), but enhanced the expression of M2 macrophage markers (CD208, Arg1, and IL10) in penile tissues [90]. They concluded that human gingiva-derived MSC treatment might be a promising strategy in treating corpus cavernosum injury-induced ED, and that human gingiva-derived MSCs exerted such functions by skewing macrophage polarity from the M1 to the M2 phenotype, subsequently protecting the corpus cavernosum from fibrosis in penile tissues [90].

As mentioned above, some of the stem cell-related anti-ED mechanisms and the relationships between stem cell-related functions and macrophages have been clarified in animal models of neurogenic ED. Endothelial progenitor cells have been reported to play crucial roles in the etiology and treatment of vascular ED via modulation of endothelial function [91]. However, there is limited information on the biological and pathological significance of macrophages in stem cell-related mechanisms in vascular ED in vivo. Regenerative therapies using various types of stem cells have been suggested as promising treatments for ED. However, there are many problems to be solved for their use in everyday practice [37,92]. Macrophages may play important roles in regenerative processes in ED. Therefore, a more detailed understanding of the pathological roles of macrophages in ED is important for the development of regenerative treatment strategies for patients with ED.

## 6. Phosphodiesterase Type 5 Inhibitors and Macrophages

### 6.1. Phosphodiesterase Type 5 Inhibitors and Macrophages in Inflammatory Diseases

PDE5 inhibitors are commonly used for patients with ED if they have no significant comorbidities, including cardiovascular diseases. Unfortunately, the pharmacological effects of macrophages in PDE5 inhibitor treatment for ED are not fully understood. On the other hand, several studies have demonstrated that PDE5 inhibitors modulate respiratory function and the status of airway diseases via regulation of inflammation and immune responses [93,94,95]. Interestingly, macrophage activation by PDE5 inhibitors played crucial roles in these findings [93,94,95]. For example, in experiments using guinea pigs, a remarkable increase in airway macrophage infiltration was observed in lipopolysaccharide (LPS)-exposed animals, which was significantly suppressed by treatment with sildenafil [93]. The study showed that sildenafil attenuated airway hyperactivity and suppression of inflammation may be associated with sildenafil-induced activity. In addition to macrophages, the LPS-induced reflux of eosinophils and neutrophils was also reduced by sildenafil treatment [93]. Therefore, the direct correlations between the influx of macrophages and sildenafil-related anti-inflammatory effects are not clear. Another report showed that vardenafil reduced the lung macrophage-induced pro-inflammatory response, including M1 polarization, which was modulated by PDE5-dependent mechanisms [95]. As these macrophages were isolated from the lungs of cystic fibrosis-model mice, the authors suggested that macrophages were the target effector cells of the anti-inflammatory effects of vardenafil [95]. The pathogenesis and progression of cystic fibrosis are regulated by many factors, such as microbial infection, cytokines, enzymes, and gene mutations [94,96,97,98]. In addition, PDE5 can affect these pathogenic activities in cystic fibrosis in various organs in a macrophage-independent manner [94,95,99,100]. Furthermore, sildenafil was reported to stimulate M2 polarization of bone marrow-derived macrophages, which was closely associated with the clearance of myelin debris and enhancement of remyelination in demyelinated tissue in a mouse model of multiple sclerosis with myelin oligodendrocyte glycoprotein-induced experimental autoimmune encephalomyelitis [101]. Thus, there is a general agreement that sildenafil can modulate macrophage activity and that it plays a role in the pharmacological mechanisms of PDE5 inhibitors.

### 6.2. Phosphodiesterase Type 5 Inhibitors and Macrophages in the Treatment

Unfortunately, there are limited reports on the relation between macrophages and PDE5 inhibitors in penile tissues [3]. In particular, the effects of PDE5 inhibitors on macrophage polarization and cytokine production in patients with ED are not fully understood. In general, PDE5 inhibitors are not used in young men, and most ED patients are older. However, PDE5 inhibitors are being used as sexual extension aids and recreational drugs in young healthy populations without medical indication [3,102]. Simsek et al. [3] investigated the histopathological and ultrastructural effects of PDE5 inhibitors in animal experiments using penile tissues from young, healthy male rats. They detected an increase in the number of fibroblasts and the diameter of the collagen fibers in the penile tissues in four-month-old-male rats treated with sildenafil, tadalafil, and vardenafil [3]. In addition, they showed that the number of macrophages in the stroma of the corpus cavernosum in all treatment groups was higher than that in the control group [3]. Unfortunately, the macrophage phenotype was not clear because this finding was obtained using electron microscopy.

As mentioned above, PDE5 inhibitors are recognized as one of the standard therapeutic agents for ED. In addition, lifestyle modifications, including physical activity, are known to be useful in improving erectile function and the efficacy of PDE5 inhibitors [56,103]. Moreover, numerous nutraceuticals and natural products have been reported to have beneficial effects and therapeutic benefits for various disorders and diseases [104,105,106]. In fact, many types of botanical medicines and natural products are reported to be useful for the prevention and treatment of ED [107,108]. For example, several animal experiments and clinical trials demonstrated that pomegranate juice improved intracavernous blood flow, smooth muscle cell relaxation, and ED symptoms [109,110,111,112]. The reported underlying mechanisms included anti-fibrotic and anti-oxidant activities modulated by NOS, NO, and malondialdehyde (MDA) in the corpus cavernosum [109,111,112]. Interestingly, an in vitro study using J774.A1 macrophages showed that the oxidative status of macrophages decreased by 56% when they were incubated with 100% pomegranate juice for 1 h [109]. In addition, the uptake of ox-LDL by macrophages was suppressed by 53% when they were treated with pomegranate juice, whereas that of non-oxidized LDL was not significantly changed [109]. Unfortunately, the study did not demonstrate the detailed pharmacological significance of such functional changes in macrophages in the improvement of erectile function.

Various male enhancement nutraceuticals are sold worldwide, and patients with ED often use these nutraceuticals [113]. However, their pharmacological mechanisms are not fully understood. In recent years, the safety, efficacy, and pharmacological quality of these nutraceutical products have been investigated using in vitro studies [113]. A variety of male enhancement nutraceuticals, such as Horny Goat Weed^®^ and AlphaMan XL^®^, suppressed LPS-induced interleukin (IL)-6 production by murine macrophages (J744 cell) in a dose-dependent manner [113]. Thus, male enhancement products are speculated to have anti-inflammatory effects, and macrophages play important roles in such pharmacological activities.

## 7. Peyronie’s Disease and Macrophages

### 7.1. Peyronie’s Disease and Erectile Dysfunction

PD is a fibrotic connective tissue disorder affecting the tunica albuginea, which is a connective tissue sheath surrounding the corpora cavernosa of the penis. As a result, fibrotic plaques develop owing to the disarranged collagen and elastin deposition [11,114,115]. In general, the initial symptom of PD is localized pain in the penis during erection. The pain eventually subsides, but penile deformation (visible curvature) is observed during erection. PD is the cause of curved-penis-related coital failure and subsequent psychosocial stress and depression [11,116,117]. Thus, PD should be regarded as not only a localized disease of the penis, but also as a systematic disease and the cause of psychosocial issues. In addition to these disorders, ED is recognized as a major symptom of PD. Despite various PD treatments, such as pharmacological, conservative, and surgical therapy [118,119,120], many investigators suggest that more innovative treatment strategies are necessary. However, the etiology and pathophysiology of PD remain unclear [115]. Therefore, in this review, we focused on the pathological roles of macrophages in PD.

### 7.2. Molecular Mechanisms Underlying Peyronie’s Disease

There is a general agreement that the initial event in the development of PD is repetitive microinjury to the tunica albuginea of the erect penis during intercourse [115,116,121]. Such traumatic events and mechanical stress are speculated to lead to an abnormal healing process in the tunica albuginea [122,123]. Various pathological processes, such as the activation of microfibrosis, persistent fibrin, increased oxidative stress, imbalance of collagen synthesis and degradation, extracellular matrix accumulation, and disorganization of elastin are triggered in the pathogenesis of PD [115,116,124]. These steps have been implicated in PD fibrosis (Figure 2). Additionally, PD fibrosis-related pathological processes are mediated by many molecules and factors, such as TGF-β1, plasminogen activator inhibitor-1, and the balance of matrix metalloproteinases (MMPs)/tissue inhibitors of metalloproteinases (TIMPs), NOS, ROS, and NO [44,114,124].

#### 7.2.1. Fibrosis and Macrophages

Fibrosis plays an important role in the cause and development of PD, and TGF-β is one of the key mediators of excessive fibrosis in PD plaques [125]. In fact, a commonly used animal model of PD is established by injecting TGF-β into the rat tunica albuginea [126,127]. Previous reports have demonstrated that TGF-β is closely associated with fibrotic processes in a multilateral manner [125]. For example, TGF-β induces fibroblast recruitment and stimulates myofibroblast proliferation and activity, resulting in increased collagen production [115,116,125,128]. Subsequently, upregulated myofibroblasts and TGF-β over-inhibit collagenolysis and fibrolysis via regulation of the turnover of the extracellular matrix, including collagen, by suppressing MMPs and stimulating TIMPs [125,129]. Furthermore, activated fibroblasts/myofibroblasts and TGF-β can modify the immune responses and oxidative stress status in PD [115,116,124,125,128]. Immune or inflammatory responses and oxidative stress also affect the etiology and pathological condition of PD plaques. Thus, myofibroblast activation and TGF-β production play central roles in the etiology of uncontrolled fibrosis in PD. In fact, some studies suggest that TGF-β is the primary pro-fibrotic factor in the formation of PD plaques [130]. Importantly, macrophages are closely associated with such TGF-β-associated processes [115,116,124]. For example, TGF-β production is significantly correlated with inducible NOS expression in the penile corpora cavernosa, and TGF-β activity is modulated by a variety of cytokines, such as monocyte chemotactic protein-1 and interferon-γ [124,131,132,133]. Immunohistochemical analyses of human samples from patients with PD showed that CD68 + cells were widely present in the peri-vascular area and on the border of the tunica albuginea, vascular plexus, and lymphoid aggregates, while CD68 + cells were rare in control samples [11]. The authors did not mention the detailed pathological significance of CD68 + cells in PD tissues. However, it is possible that macrophages play a crucial role in the pathological conditions of PD.

#### 7.2.2. Other Factors and Future Directions

In this review, we summarized the pathogenesis of PD in a way that is easy to understand for general urologists and investigators who are not PD experts. In addition to what we discussed in this review, many genetic, molecular, and systemic factors, such as abnormalities of chromosomes 7 and 8, interleukins, phosphoinositide 3-kinase/Akt signaling, Rho/ROCK pathway, Smads, and diabetes mellitus, are known to play crucial roles in the development of PD via complex mechanisms [115,125,134,135,136,137]. Therefore, we emphasize that the readers should also refer to previous well-written reviews on PD pathogenesis [11,115,124,125,137].

Various novel treatment strategies and agents have been suggested and used for PD [124,138,139,140,141]. Furthermore, several studies have analyzed the usefulness of treatments according to disease phases [140,141,142]. However, the study populations in most pre-clinical and clinical studies vary widely in disease duration from the disease onset, and molecular mechanisms under pathological conditions are dependent on the disease phase. Additionally, there are no standard therapies with secure effects. Therefore, future therapeutic studies using various new technologies, such as next-generation sequencing and phenotypical screening assays, are required [115]. Moreover, the regulation of macrophage activities may play pharmacological roles in novel treatment strategies.

## 8. Conclusions

In this review, we discussed the pathological significance and pharmacological roles of macrophages in ED. Many factors affect the etiology and pathological conditions of ED via complex mechanisms. In addition to PDE5 inhibitors, lifestyle modification, PE, and various agents, including supplements, are known to be useful in the treatment of ED. Although their pharmacological roles remain unclear, macrophages may play a crucial role in such anti-ED actions. PD is a recognized cause of ED, but the relevant underlying pathological mechanisms remain unclear at the molecular level, and treatment strategies are not satisfactory. Importantly, macrophages play a significant role in various ED aspects by regulating endothelial function, nerve injury repair, cytokine production, and fibrosis. Therefore, understanding the activities of macrophages is essential to discussing ED etiology, prevention, observation, and treatment. However, the detailed pathological and pharmacological roles of macrophages in ED, including PD, are not fully understood, especially in human penile tissues. This is exemplified by the fact that there are no clinical trials regarding the use of novel agents targeting macrophages for the treatment of ED or Peyronie’s disease. Although there are ethical issues regarding the use of human penile tissues, further in vivo studies are necessary to understand the activities of macrophages in ED. Finally, clinical trials that target macrophages are greatly anticipated.

## Figures and Tables

**Figure 1 biomedicines-09-01658-f001:**
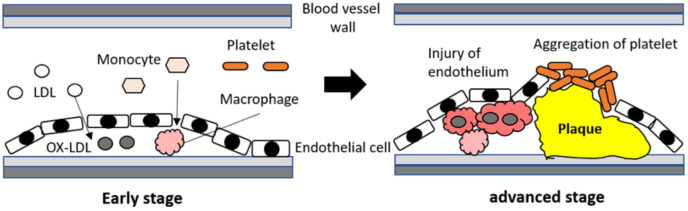
Summary of the pathological roles of oxidized low-density lipoprotein and macrophages in endothelial dysfunction and vascular damage.

**Figure 2 biomedicines-09-01658-f002:**
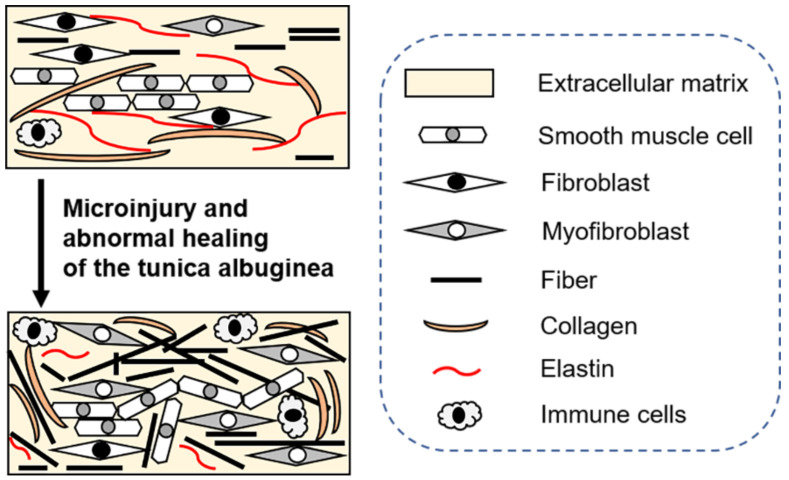
Schematic of the etiology and development of Peyronie’s disease.

**Table 1 biomedicines-09-01658-t001:** Summary of pathological roles of macrophages in erectile dysfunction.

Cause of ED	Pathological Roles	References
Vasculogenic	Endothelial injury and plaque instability due to endothelial dysfunction	[32,33]
	Reactive oxygen species production and cytokine secretion	[8,35,36]
	Modulation of lipid metabolism (oxidization of low-density lipoprotein)	[8]
Neurogenic	Modulation of cavernous nerve injury and nitrergic nerve apoptosis	[42,43]
	Impairment of erectile tissue physiology	[42]
	Regulation of corporal smooth muscle function	[43]
Obesity-related	Increase in TNF-α expression in the corpus cavernosum	[55,56]
	Stimulation of inflammatory status in penile tissues	[56]

ED, erectile dysfunction; TNF, tumor necrosis factor.

## Data Availability

Not applicable.

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
