# Peer review of "Pathological Significance of Macrophages in Erectile Dysfunction Including Peyronie’s Disease"

_biomedicines, 2021, doi:10.3390/biomedicines9111658_

Round 1

Reviewer 1 Report

The authors describes pathological significance of macrophages in erectile dysfunction including Peyronie’s disease. This review article is written from unique, but scientifically-solid views on important issues of erectile dysfunction. Since recent studies have implicated the importance of macrophage in the pathology and treatment of erectile dysfunction, I think this review article will gain broad interests by readers in the future.

I would like to point out one advice. The authors should show a table (or a figure) to summarize the functions of macrophage in the physiology of erectile function and the pathology of erectile dysfunction. 

Author Response

Reviewer 1

The authors describes pathological significance of macrophages in erectile dysfunction including Peyronie’s disease. This review article is written from unique, but scientifically-solid views on important issues of erectile dysfunction. Since recent studies have implicated the importance of macrophage in the pathology and treatment of erectile dysfunction, I think this review article will gain broad interests by readers in the future.

I would like to point out one advice. The authors should show a table (or a figure) to summarize the functions of macrophage in the physiology of erectile function and the pathology of erectile dysfunction. 

(Response)

   We thank the reviewer for evaluating our manuscript. We agree with the comment, and the reviewer’s suggestion has helped to improve the manuscript considerably. Based on this suggestion, we added a new table (Table 1) regarding the pathological roles of macrophages in ED. Our revisions are indicated in red font within the marked version of the manuscript.

Moreover, the revised manuscript has also been proofread by a native speaker according to the editor’s comment.

Reviewer 2 Report

This is a well conducted review article regarding the important pathological significance and pharmacological roles of macrophages in ED and Peyronie's disease. And the information in this manuscript is useful and of clinical importance.

minor comment:

1: Is there any clinical trial using novel agents targeting macrophage for treatment ED or Peyronie’s disease?

Author Response

Reviewer 2

This is a well conducted review article regarding the important pathological significance and pharmacological roles of macrophages in ED and Peyronie's disease. And the information in this manuscript is useful and of clinical importance.

minor comment:

1: Is there any clinical trial using novel agents targeting macrophage for treatment ED or Peyronie’s disease?

(Response)

   We thank the reviewer for evaluating our manuscript and the important question. We again searched for clinical trials using novel agent targeting macrophages for treatment of ED or Peyronie’s disease. Unfortunately, there were no such clinical trials. We think this information is important to understand the current status of the clinical significance of macrophages in ED or Peyronie’s disease. Therefore, we added this information to section 8. Conclusions.

Moreover, the revised manuscript has also been proofread by a native speaker according to the editor’s comment.
